# Effects of *Porphyra tenera* Supplementation on the Immune System: A Randomized, Double-Blind, and Placebo-Controlled Clinical Trial

**DOI:** 10.3390/nu12061642

**Published:** 2020-06-02

**Authors:** Su-Jin Jung, Hui-Yeon Jang, Eun-Soo Jung, Soon-Ok Noh, Sang-Wook Shin, Ki-Chan Ha, Hyang-Im Baek, Byung-Jae Ahn, Tae-Hwan Oh, Soo-Wan Chae

**Affiliations:** 1Clinical Trial Center for Functional Foods, Chonbuk National University Hospital, Jeonju 54907, Korea; sjjeong@jbctc.org (S.-J.J.); janghy@jbctc.org (H.-Y.J.); esjung@jbctc.org (E.-S.J.); sono@jbctc.org (S.-O.N.); 2Department of Medical Nutrition Therapy, Jeonbuk National University, Jeonju 54896, Korea; tivet21@gmail.com; 3Healthcare Claims & Management Incorporation, Jeonju 54858, Korea; omphalos9121@hanmail.net (K.-C.H.); hyangim100@gmail.com (H.-I.B.); 4Marine Biotechnology Research Center, Jeonnam Bioindustry Foundation, Wando 59108, Korea; chemeditech@hanmail.net (B.-J.A.); sosoth2@hanmail.net (T.-H.O.)

**Keywords:** *Porphyra tenera*, immune, clinical trial, natural killer cells, cytokines

## Abstract

Objective: The purpose of this study was to determine if *Porphyra tenera* extract (PTE) has immune-enhancing effects and is safe in healthy adults. Methods: Subjects who met the inclusion criteria (3 × 10^3^ ≤ peripheral blood leukocyte level ≥ 8 × 10^3^ cells/µL) were recruited for this study. Enrolled subjects (*n* = 120) were randomly assigned to either the PTE group (*n* = 60) and were given 2.5 g/day of PTE (as PTE) in capsule form or the placebo group (*n* = 60) and were given crystal cellulose capsules with the identical appearance, weight, and flavor as the PTE capsules for 8 weeks. Outcomes were assessed based on measuring natural killer (NK) cell activity, cytokines level, and upper respiratory infection (URI), and safety parameters were assessed at baseline and 8 weeks. Results: Compared with baseline, NK cell activity (%) increased for all effector cell-to-target cell ratios in the PTE group after 8 weeks; however, changes were not observed in the placebo group (*p* < 0.10). Subgroup analysis of 101 subjects without URI showed that NK cell activity in the PTE group tended to increase for all effector cell/target cell (E:T) ratios (E:T = 12.5:1 *p* = 0.068; E:T = 25:1 *p* = 0.036; E:T = 50:1 *p* = 0.081) compared with the placebo group. A significant difference between the two groups was observed for the E:T = 25:1 ratio, which increased from 20.3 ± 12.0% at baseline to 23.2 ± 12.4% after 8 weeks in the PTE group (*p* = 0.036). A significant difference was not observed in cytokine between the two groups. Conclusion: PTE supplementation appears to enhance immune function by improving NK cell activity without adverse effects in healthy adults.

## 1. Introduction

Immune system imbalances can be caused by various factors, including aging, viral disease, increased stress, and environmental pollution. The interest in functional foods and alternative treatments to improve immunity has increased because pharmacologically active substances in foods may boost the immune system. Mass production of sea algae using aquaculture technology has been used to produce value-added foods [1,2]. Laver (*Porphyra tenera*), a type of red algae, has long been a staple food eaten in Western Pacific Ocean regions, including South Korea, China, and Japan, and its consumption is steadily increasing not only in Southeast Asia but also around the world. Laver is low in calories but rich in carbohydrates (CHO), protein, vitamins, and minerals. Unlike other sea algae substances, laver has many free sugars, such as isofloridoside and floridoside, which are major carbohydrates. In addition, laver has abundant dietary fiber in the form of sea algae polysaccharides including hemicellulose, which is an insoluble polysaccharide and cell wall component, and porphyrans, which are water-soluble acidic polysaccharides and intercellular rechargeable substances [3,4]. The porphyran present in laver comprises 3,6-anhydro-L-galactose, D-galactose, ester sulfate, and 6-O-methyl-D-galactose components. This porphyran is not digestible by humans and is highly viscous; thus, it can be used as a diluent agent [5,6]. Fucoidan is another component of edible sea algae (green or brown algae) and considered a functional food, food additive, and physiologically active substance [7,8] The interest in laver as a functional food has increased, particularly in the antioxidant activities of porphyrans and polyphenols in laver [9,10]. Laver has been reported to degrade cholesterol [11], show anti-tumor and immune enhancement activities [12], function as an antioxidant [9,13], improve lipid metabolism [14], have anti-inflammatory activities [15], and exert anti-mutagenic effects [16]. In in vitro and in vivo studies, *Porphyra tenera* extract (PTE) administration was shown to increase survival rates of splenocytes and macrophages; increase NF-kB stimulation; increase NO production; increase levels of the Th1 cytokines interleukin-2 (IL-2), IL-12, interferon gamma (IFN-γ), and tumor necrosis factor alpha (TNF-α); and increase immune function by promoting iNOS secretion [17,18]. However, the immune enhancement effects of PTE have not been verified in clinical trials. In the present study, the effects of PTE supplementation on immunological indicators in healthy adults were examined, and its safety was investigated.

## 2. Materials and Methods

### 2.1. Test Supplements

Ethanol extraction of 100 kg dried laver (*Porphyra tenera*) (Wando, Jeollanamdo Province, Korea) was performed at 80 ± 2 °C for 3 h using 10% ethanol. PTE was obtained using a lyophilization method after filtration using a 1 µm housing filter, and the extract was concentrated to approximately 10–20 Brix at 65–70 °C. The yield rate of PTE was 13%, and the concentration of porphyra 334 in this PTE was 68.45 ± 20% mg/g. An HPLC chromatogram of the PTE is shown in Figure 1. Based on our preclinical studies [17,18], administering oral PTE doses of 500 and 1000 mg/kg to 6-week-old ICR mice significantly improved immunity-related indicators by promoting the secretion of cytokines and iNOS. This is equivalent to a human daily intake of 2.5 g per day based on a dose of 500 mg/kg. All test products used in this study were provided by the Marine Biotechnology Research Center (Wando, Jeollanamdo Province, Korea) in the form of yellow-brown capsules (powder). Placebo products were made of crystal cellulose and had the same appearance, flavor, and weight as the test products (Table 1).

### 2.2. Subjects

Subjects were recruited from 22 May to 22 August 2019 through internal advertising (Internet postings on departmental home pages, brochures, and posters) at the Clinical Trial Center for Functional Foods (CTCF2) at Chonbuk National University Hospital. This study was conducted after receiving approval from the Institutional Review Board (IRB) of Chonbuk National University Hospital (IRB No. CUH 2019-04-050), and the protocol was registered at www.clinicaltrials.gov (NCT04017988). The entire study was performed in accordance with the Helsinki Declaration and the provisions of the Korean Good Clinical Practice (KGCP). A total of 120 participants was eligible after screening tests such as questionnaires, physical examinations, and laboratory examinations and were enrolled within 3 weeks after providing informed consent and before given supplement. Selection criteria were as follows:(1)Males and females >50 years of age at the time of the screening test;(2)Written consent to participate provided prior to the start of the study;(3)A peripheral white blood cell (WBC) count > 3 × 10^3^ and < 8 × 10^3^ cells/μL as measured in the screening test.

Exclusion criteria were as follows:(1)Vaccination against influenza within 3 months prior to the screening test;(2)Body mass index (BMI) < 18.5 kg/m^2^ or > 35 kg/m^2^ at the time of the screening test;(3)Presence of a clinically acute disease or chronic cardiovascular, endocrine, immune, respiratory, hepatobiliary, kidney, urinary, neuropsychiatric, musculoskeletal, inflammatory, hematological, or gastrointestinal disease;(4)Supplementation with medicines or health functional foods associated with immunity enhancement within 1 month prior to the screening test;(5)Administration of antipsychotics within 3 months prior to the screening test;(6)Suspected alcoholism or drug abuse;(7)Participation in other human tests within 3 months prior to the screening test;(8)The following diagnostic and medical test results:☞ aspartate transaminase (AST) or alanine transaminase (ALT) > 3x the normal upper limit.☞ Serum creatinine >2.0 mg/dL.
(9)Pregnant or nursing;(10)Those who were fertile and not taking contraceptives;(11)Deemed inappropriate to participate in the research for other reasons, including the results of diagnostic and medical examinations.

### 2.3. Study Design

The study was a randomized, double-blind, placebo-controlled study to assess the effectiveness and safety of PTE at enhancing immune function when taken for 8 weeks. A total of 120 subjects participated in the study, with 60 individuals each assigned to the test and placebo groups. Subjects who met the selection criteria were assigned to the test group or placebo group based on an allocation code generated using a block-based random assignment method. Subjects took the test capsule or placebo capsule twice a day (2 capsules at a time) for 8 weeks after breakfast and dinner.

Screening was conducted once for all subjects, and they received a basic evaluation (baseline) on the day of the first visit (Week 0). After the first visit, subjects visited CTCF2 every 4 weeks (2nd visit: Week 4, 3rd visit: Week 8) to examine vital signs, drug dosage, medical condition changes, and adverse reactions. Intake of the test capsules was examined and monitored by the test manager by directly counting the number of remaining capsules to monitor compliance (intake rate, %) at the second visit (Week 4) and third visit (Week 8).

### 2.4. Randomization

In the present study, the same block size was applied for balanced random assignment in the intake groups, and the number of assigned subjects in each group was determined using a 1:1 ratio. A random assignment table was generated using the randomization program in SAS^®^ from subject number 1 using random number permutations (random numbers of A and B). When packing the test capsules, the test capsule labels were attached based on the random assignment table and supplied to CTCF2 before starting the study. Allocation codes were managed by the study investigators and were monitored by clinical research associates until the end of the study. During the research period, all research investigators and participants were blinded to identification codes.

### 2.5. Outcome Measurements

All subjects who visited CTCF2 underwent a safety assessment at the first visit (baseline Week 0) and third visit (Week 8).

#### 2.5.1. Clinical and Biochemical Analyses

Hematology examination, blood biochemical tests, and urine tests were performed in blood and urine collected after a 12-h fast.

#### 2.5.2. Primary Outcomes

NK cell activity was assessed using the CytoTox 96W Non-Radioactive Cytotoxicity Assay kit (Promega Corp., Madison, WI, USA). After separating peripheral blood mononuclear cells (PBMCs) from heparin-treated venous blood using a density gradient centrifugation method, the cell-mediated cytotoxicity of NK cells in peripheral blood monocytes was measured [19]. The lactate dehydrogenase (LDH) assay method [20] was used with target cells of K562 cells (human leukemia cell line, Korean human leukemia cell line bank, Seoul, Korea). Effector cell/target cell (E:T) ratios were 50:1, 25:1, and 12.5:1. Cytotoxicity (%) was calculated as described below by measuring both natural release and maximum release at the first visit (baseline Week 0) and third visit (Week 8):

Cytotoxicity (%) = (experimental release − effector spontaneous release − target spontaneous release)/(target maximum release − target spontaneous release) × 100

#### 2.5.3. Secondary Outcomes

Cytokines (IL-2, IL-6, IL-12, interferon-γ, and TNF-α) were measured at the first visit (baseline Week 0) and third visits (Week 8). The incidence of upper respiratory infection (URI) was assessed on the day of the visit for the screening test (1st visit, Week 0), the second visit (Week 4), and the third visit (Week 8). To analyze blood cytokines, 3 mL of blood was collected in a 5-mL SST tube (Hanil Science Industrial Co. Ltd., Seoul, Korea) and centrifuged for 10 min at 3000 rpm (or 2000× *g*) after sitting at room temperature for 30 min to clot. Approximately 1 mL of supernatant was transferred to a microtube and kept frozen at −70 °C until analysis. Serum IFN-γ, IL-2, IL-6, IL-12, and TNF-α levels were assessed using the enzyme-linked immunosorbent assay (ELISA) kit (VersaMax Microplate Reader, Molecular Devices, San Jose, CA, USA). The incidence and frequency of URIs were evaluated at the first visit (Week 0), the second visit (Week 4), and the third visit (Week 8) using the URI symptom assessment tool (Wisconsin Upper Respiratory Symptom Survey (WURSS) [21]). Symptom scores and duration (days) were also recorded. Examination items were signs of URI, presence or absence of URI symptoms, and types of symptoms (throat pain, sniffing, nasal congestion, sneezing, hoarseness, myalgia, earache, pyrexia, headache, coughing, phlegm, labored breathing, diarrhea, nausea, and vomiting). The severity of symptoms was assessed as follows: 0 for no sign of symptoms, 1 for mild symptoms, 2 for average symptoms, and 3 for serious symptoms.

### 2.6. Safety Outcome Measurements

Clinical conditions of the subjects including adverse reactions were evaluated and recorded in the case report list. All subjects underwent safety evaluations at baseline (Week 0) and after completing the 8-week study period. Safety assessments included electrocardiograms, vital signs (blood pressure and pulse rate), and laboratory tests. In the hematological examination, WBC count, red blood cell (RBC) count, hemoglobin level, hematocrit, platelet, neutrophil, lymphocyte, and basophil counts were examined. In the blood biochemical examination, liver and kidney functions were investigated including total bilirubin, total protein, alkaline phosphatase (ALP), gamma-glutamyl transferase (gamma-GT), ALT, AST, blood urea nitrogen (BUN), glucose, and creatinine assessment. Levels of lipid metabolic indicators of total cholesterol, triglycerides, high-density lipoprotein cholesterol (HDL-C), and low-density lipoprotein cholesterol (LDL-C) were measured.

### 2.7. Evaluation of Diet and Physical Activity

To investigate changes in dietary habits, a nutritionist trained in the dietary records method explained to subjects how to prepare a dietary diary and collected data after confirmation through face-to-face interviews when retrieving the dietary diary. During the first visit (baseline; Week 0) and the third visit (Week 8), the dietary diary from the previous day was collected and analyzed using CAN-Pro 4.0^®^, a computer-aided nutritional analysis program of the Korean Nutrition Society Forum, Seoul, Korea, and average values were calculated. To investigate changes in physical activity, the global physical activity questionnaire (GPAQ) [22] was administered on the first visit (baseline; Week 0) and the third visit (Week 8), and the metabolic equivalent task (MET) value was calculated using the data. Subjects were instructed to maintain their regular daily lifestyle, diet, and physical activity during the study.

### 2.8. Statistical Analysis

All statistical analyses were performed using SAS^®^ version 9.4 (SAS Institute, Cary, NC, USA). All data are presented as mean ± standard deviation (SD) for continuous variables and as n (%) for categorical variables. Final results were evaluated using full analysis set (FAS) and per protocol (PP) analysis. PTE and placebo groups were compared using independent *t*-tests for continuous variables collected at baseline, while categorical variables and URI incidence rates were compared using the chi-square test (Fisher’s exact test). Changes in NK cell activity and cytokine levels as continuous variables before and after intake were analyzed using paired *t*-test, and differences between intake groups were analyzed using a linear mixed model. In several previous studies, URIs reportedly changed the activity of cytokines and NK cells [23,24,25]. Therefore, subgroup analysis was conducted based on the incidence of URI. Baseline and demographic variables for heterogeneous evaluation items were calibrated with covariates for analysis of covariance (ANCOVA) testing. A *p*-value < 0.05 was considered statistically significant.

### 2.9. Sample Size

The primary outcome in this study was difference in NK cell activity between the PTE and placebo groups after 8 weeks of supplement intake. A power calculation was applied based on the results and design of a previous study [26]. The change in NK cell activity in the PTE group was assumed to be 7.4%, and the changes in the placebo group and SD were assumed to be −0.28% and 12.5%, respectively. The sample size required to maintain 80% statistical power at a 5% significance level (two-tailed test) was calculated to be 42 persons per group. Therefore, a total of 120 people (60 subjects per group) was required assuming a dropout ratio of 30%.

## 3. Results

### 3.1. Participant Demographic Characteristics

After screening a total of 137 volunteers, 120 were selected (20 males, 100 females) as eligible candidates and randomly assigned to the PTE or placebo group (*n* = 60 each group). A total of 111 subjects completed the study protocol. Nine subjects were excluded due to drop-out or violation of the study plan (2 violators of the exclusion criteria in the PTE group, 2 violators of the study plan in the placebo group, and 5 consent withdrawals after the first visit in the placebo group). Data from 111 subjects were used for analyses (Figure 2). Significant differences were not observed in compliance with regard to taking the supplements between the PTE group and placebo group (95.2 ± 5.8% in the PTE group vs. 96.2 ± 4.1% in the placebo group, *p* = 0.265). In addition, significant differences were not observed in age, sex, BMI, vital signs, smoking rate, smoking quantity, drinking rate, or drinking amount between the two groups (*p* > 0.05; Table 2). Although significant differences were not observed in NK cell activity or IL-2, IL-6, IL-12, and IFN-γ levels between the two groups, TNF-α was significantly higher at baseline in the placebo group than in the PTE group (*p* = 0.017; Table 2). Therefore, statistical results were calibrated for this difference in baseline TNF-α.

### 3.2. Dietary Intake and Physical Activity

Dietary intake results are presented in Table 3. Carbohydrate (CHO) intake in the PTE group increased significantly after 8 weeks (*p* = 0.002) compared with baseline, leading to a significant difference in CHO intake between the two groups at 8 weeks (*p* = 0.0198). Statistically significant differences were not observed in calorie, protein, fat, or fiber intake between the two groups (*p* > 0.05). Results are presented after calibrating for CHO intake. Significant changes were not found in MET within or between groups (*p* > 0.05).

### 3.3. Efficacy Evaluation

#### 3.3.1. Primary Outcome

Changes in NK cell activity level in the PTE and placebo groups are presented in Table 4. NK cell activity (%) increased in the PTE group for all E:T ratios (E:T = 12.5:1 *p* = 0.0004, E:T = 25:1 *p* = 0.0034, and E:T = 50:1 *p* = 0.0055) compared with baseline but did not increase in the placebo group. Comparison of changes in NK cell activity between the placebo and PTE groups after adjusting for baseline carbohydrate and MET showed an increased tendency in the PTE group compared with the placebo group (*p* < 0.10) for all E:T ratios (E:T = 12.5:1 *p* = 0.0829, E:T = 25:1 *p* = 0.0587, and E:T = 50:1 *p* = 0.0832). NK cell activity levels in individuals who did not have a URI (*n* = 101) are presented in Table 5. NK cell activity in the PTE group after 8 weeks was increased compared with baseline for all E:T ratios (E:T = 12.5:1 *p* = 0.0019, E:T = 25:1 *p* = 0.0106, and E:T = 50:1 *p* = 0.0134), but no such increase was noted in the placebo group. NK cell activity in the PTE group showed a greater tendency to increase compared with the placebo group (*p* < 0.10) for all E:T ratios (E:T = 12.5:1 *p* = 0.0683, E:T = 25:1 *p* = 0.0357, and E:T = 50:1 *p* = 0.0810). NK cell activity in the PTE group at the ratio of E:T = 25:1 increased significantly to 23.2 ± 12.4% after 8 weeks compared with 20.3 ± 12.0% at baseline (*p* = 0.0357).

#### 3.3.2. Secondary Outcomes

Cytokine concentrations in the PTE and placebo groups are presented in Table 4. Significant differences were not observed in serum IL-2, IL-6, IL-12, or IFN-γ between baseline and after 8 weeks of supplementation in either the PTE or placebo groups for the full analysis set (*n* = 111) or in cytokine levels between the two groups (*p* > 0.05). Although the TNF-α concentration in the PTE group appeared to increase after 8 weeks compared with baseline, the increase was not statistically significant in the PTE group or significantly different between the two groups (*p* > 0.05). Other cytokines showed no significant differences between the two groups (*p* > 0.05). Sub-analysis of cytokine concentrations in subjects without a URI (*n* = 101) are presented in Table 5. Comparison of IL-2, IL-6, IL-12, and IFN-γ levels at baseline and after 8 weeks of supplementation showed no significant differences within or between groups (*p* > 0.05). TNF-α level in the PTE group significantly increased (*p* = 0.0368) after 8 weeks compared with baseline. Significant differences were not observed in cytokine levels between the placebo and PTE groups (*p* > 0.05). Among all subjects (*n* = 111), URI in the PTE group decreased by 6.9% and 3.5% from baseline after 4 and 8 weeks of supplementation, respectively; however, the differences were not statistically significant (*p* > 0.05; Appendix A).

### 3.4. Safety and Adverse Events

Subjects in this study showed no significant changes or differences in safety indicators such as laboratory tests, electrocardiograms, or vital signs (BP and pulse) during the study (*p* > 0.05). All laboratory test items were within the normal range, and no side-effects were observed. Regarding adverse reactions, 11 slight or severe abnormalities occurred in 11 subjects among the 120 subjects; 8 abnormalities were in the PTE group and 3 in the placebo group; however, this difference was not statistically significant (*p* > 0.05; Appendix A). Adverse reactions included 1 case of abdominal discomfort, 4 cases of heartburn, 1 case of contact dermatitis (left leg), 1 case of left knee pain, 1 case of chronic dermatitis (leg) deterioration, 1 case of left trigger finger, 1 case of increased liver enzyme ratio (AST/ALT), and 1 case of burn (back of the left hand). Five of these cases did not have a clear causal relationship between the adverse reaction and intake of the test product and were therefore not considered relevant; however, 6 cases could have been caused by supplement intake. 

## 4. Discussion

The purpose of this study was to evaluate the effectiveness and safety of PTE supplementation in enhancing immune function in healthy adults. To the best of our knowledge, this is the first randomized, double-blind, and placebo-controlled study in which the immune-boosting effects of laver were determined. Supplementation with 2.5 g of PTE per day for 8 weeks had beneficial immune regulation effects without adverse effects. Based on several previous studies [7,24,25,27], influenza vaccination and influenza virus infection increase NK cell activity and antibody titers, which significantly affect immune parameters. Thus, in the present study, subgroup analysis of subjects without a URI was performed to exclude the effects of viral infection on NK cell activity. In subgroup analysis of subjects without a URI (*n* = 101) and of all subjects (*n* = 111), NK cell activity tended to increase in the PTE group for all E:T ratios (E:T = 12.5:1, E:T = 25:1, and E:T = 50:1; *p* < 0.10) compared with the placebo group; however, significant differences were not observed between the two groups. Subjects without a URI in the PTE group showed increased NK cell activity at an E:T ratio of 25.1:1 compared with the placebo group. In several previous studies [19,26,28,29,30,31,32,33], enhanced NK cell activity in healthy adults was reported only at certain E:T ratios; in the present study, enhanced NK cell activity was consistently identified in the PTE group at all E:T ratios (E:T = 12.5:1, E:T = 25:1, and E:T = 50:1).

In general, active Th cells differentiate into two functional subclasses, Th1 and Th2, which are categorized based on the types of cytokines they secrete. Th1 cells produce Th1 cytokines (pro-inflammatory cytokines) such as IFN-γ, TNF-α, IL-2, and IL-12 that improve cellular immunity against intracellular pathogens such as bacteria. Th2 cells produce Th2 cytokines (anti-inflammatory cytokines) such as IL-4, IL-5, IL-6, and IL-10 that are involved in combating extracellular parasite infections and contributing to humoral immunity [34]. Th1 cytokines affect immune responses and control NK cell activity [35]. When NK cells are activated, they secrete various cytokines such as IFN-γ and TNF-α and effectively exterminate target cells [36]. In preclinical studies [17,18], oral intake of PTE for 4 weeks increased the secretion of cytokines (IL-1β, IL-2, IL-4, and IFN-γ) by spleen cells and promoted iNOS expression in ICR mice. In addition, PTE stimulated T-helper cell type immune reactions and promoted the secretion of cytokines (IL-10, IL-6, TNF-α, and IFN-γ) in RAW264.7 cells and mouse splenocyte and macrophages cells, indicating that PTE has immune regulatory effects. Consistent with a previous animal study in which TNF-α generation by macrophages in response to PTE was increased [18], an increase in TNF-α level was observed in the present study after 8 weeks of PTE supplementation compared with baseline. These results are partially consistent with the study by Jiang et al. [37] who showed that the porphyran fraction of PTE inhibited NO generation in macrophages by blocking NF-kB activation [17] in macrophages. Taken together, the results of previous studies indicate that the porphyran component of PTE likely stimulates Th1 cells and improves cell-mediated immunity, as reflected by increased NK cell activity with generation of the Th1 cytokine TNF-α. However, Th-2 induced cytokines (IL-4, IL-6, and IL-10) were not measured in the present study. Kwak et al. [28] reported that changes in NK cell activity were associated with serum IFN-γ level. The changes in NK-cell activity and cytokines identified in the present study are consistent with the results reported in several earlier studies [26,31,32] and case reports [19,28,30,38,39]. Healthy adults who received *Cordyceps militaris* [31] supplementation for 8 weeks or a mycelium extract of *Cordyceps* [32] for 8 weeks showed an increase in NK cell activity without a change in cytokine levels, consistent with the findings in the current study. *Cordyceps militaris* increased IL-2 and IFN-γ levels in addition to NK-cell activity in healthy males (average age, 36.5 ± 11.2 years) after 4 weeks of supplementation [38]; however, after 12 weeks of intake, only NK cell activity was increased in healthy males and females (average age 48.9 ± 6.7 years [31]), similar to the results in the present study. In other studies, intake of *Chlorella* [28] by healthy adults (WBC counts 4000–8000 cells/μL) for 8 weeks increased cytokine levels (IFN-γ, interleukin-1β) as well as NK cell activity. Furthermore, supplementation with silk peptide [30] in non-seasonal influenza vaccine subjects for 8 weeks increased levels of cytokines (IL-2, IL-12, and IFN-γ) and NK cell activity. Intake of tulsi (*Ocimum sanctum* Linn.) leaf extract [39] for 4 weeks only increased cytokine levels (IFN-γ, IL-4); however, intake of β-1,3-glucan [19] by adults with severe stress levels increased the IL-10 level as well as NK cell activity. The results of the previous studies differ from those of the current study. In the studies mentioned above, both NK cell activity and cytokine levels changed with supplementation in healthy adults under 40 years of age [19,28,29,33,38,39]. Factors that stimulated the increase in NK cell activity may also have stimulated an increase in IFN-γ expression through different mechanisms. As mentioned above, the differences in cytokine activity among the studies may be due to various factors such as age [19,26,28,29,31,32,33,38,39,40], inherent immunity, WBC count [28,29,30,31,32], availability of influenza vaccination [24,27], seasonal factors (URI preventive inoculation season [30]), intake period of test supplements [29,30,31,32,33,38], and stress [19]. In the current study, the URI number among all 111 participants was 10 (*n* = 8 in the PTE group and *n* = 2 in the placebo group); the difference was not statistically significant. Because NK cell activity is the strongest predictor of immune function [25,41], weakening or loss of NK cell activity may be a sign of health issues. Clinically meaningful adverse reactions or body changes were not observed with PTE supplementation in the present study, indicating that PTE is safe to use in humans.

The present study had several strengths. The effects of URIs and exposure to influenza infection were minimized as much as possible to diminish the effects of URI and influenza on immune parameters (NK cell activity, cytokines, and antibody titers [7,24,25,27]) during the research period. Specifically, after recruiting all subjects within a short period of time (1 month), seasonal factors and individual variations were minimized by starting and completing the research in 3 months (22 May–22 August 2019), which allowed assessment of whether PTE can enhance immune function in subjects without a URI by improving NK cell activity.

The present study had several limitations. PTE did not change the expression of immune cytokines, indicating that it may have increased NK cell activity by facilitating generation of the Th1 cytokine, TNF-α. Second, future research should include a larger study cohort to confirm this finding. Third, immune cytokine levels were evaluated in the absence of trigger factors (e.g., stress, fatigue, exercise) that stimulate immune responses in healthy middle-aged subjects with no specific diseases. The exact effects of PTE on immune function should be investigated in future studies with consideration of the age at which immunity decreases, selecting subjects with high stress levels, and conducting the study outside of the influenza vaccination season.

## 5. Conclusions

PTE supplementation appears to enhance immune function by improving NK cell activity without adverse effects in healthy adults. During the study period, clinically significant changes or adverse effects were not observed as assessed based on laboratory tests, ECG, and vital signs. Together, these results indicate that 8 weeks of PTE supplementation can improve immune function by improving NK cell activity in healthy subjects.

## Figures and Tables

**Figure 1 nutrients-12-01642-f001:**
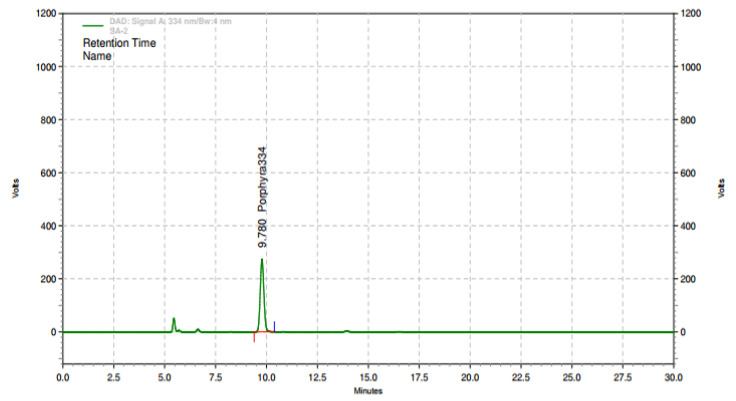
Representative chromatograms of *Porphyra tenera* extract (PTE) based on high-performance liquid chromatography (HPLC) analysis of porphyra34 in PTE.

**Figure 2 nutrients-12-01642-f002:**
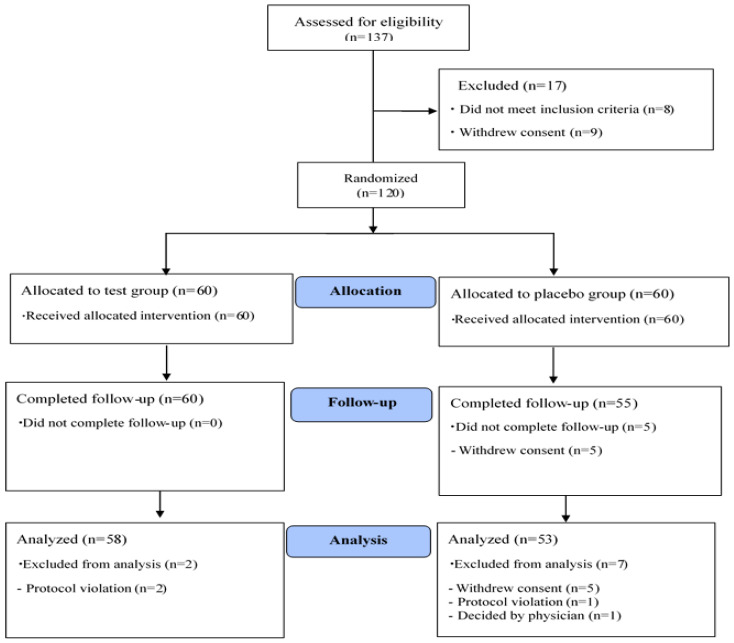
Flow chart of study participants.

**Table 1 nutrients-12-01642-t001:** Composition of the test product as determined by HPLC.

Component	Test Capsule PTE Supplement (%)	Placebo Supplement (%)
PTE	99.52	-
Silicon dioxide	0.48	0.2
Microcrystalline cellulose	-	99.6
Caramel coloring	-	0.2
Total	100	100

HPLC, high-performance liquid chromatography; PTE, *Porphyra tenera* extract.

**Table 2 nutrients-12-01642-t002:** Baseline general characteristics of the subjects.

	PTE Group	Placebo Group	Total	*p*-Value ^1^
	(*n* = 58)	(*n* = 53)	(*n* = 111)	
Sex (M/F) (*n*.%)	9(15.5)/49(84.5)	10(18.9)/43(81.1)	19(17.1)/92(82.9)	0.640 ^2^
Age (years)	59.6 ± 4.8	59.9 ± 6.1	59.8 ± 5.5	0.718
Height (cm)	158.3 ± 6.5	157.9 ± 7.5	158.1 ± 7.0	0.803
Weight (kg)	60.1 ± 8.1	59.3 ± 10.1	59.7 ± 9.1	0.651
BMI (kg/m^2^)	23.9 ± 2.7	23.7 ± 3.1	23.8 ± 2.8	0.615
SBP (mmHg)	120.7 ± 15.6	120.1 ± 17.7	120.4 ± 16.6	0.852
DBP (mmHg)	70.4 ± 10.8	69.7 ± 11.9	70.1 ± 11.3	0.740
Pulse (bpm)	69.3 ± 9.6	69.6 ± 9.2	69.5 ± 9.4	0.869
NK cell activity 12.5:1 (%)	11.4 ± 8.4	11.8 ± 9.1	11.6 ± 8.7	0.828
25:1 (%)	20.6 ± 12.2	20.7 ± 13.6	20.6 ± 12.8	0.992
50:1 (%)	28.0 ± 14.3	27.4 ± 14.3	27.7 ± 14.3	0.837
IL-2 (pg/mL)	1.28 ± 0.85	1.44 ± 1.51	1.36 ± 1.21	0.462
IL-6 (pg/mL)	0.64 ± 0.69	0.63 ± 0.85	0.63 ± 0.77	0.982
IL-12 (pg/mL)	2.31 ± 1.86	2.08 ± 1.49	2.20 ± 1.69	0.466
INF-γ (pg/mL)	8.35 ± 8.68	7.04 ± 6.97	7.72 ± 7.90	0.384
TNF-α (pg/mL)	6.05 ± 2.47	7.36 ± 2.82	6.68 ± 2.71	0.011
Alcohol (n, %)	23(39.7)/35(60.3)	27(50.9)/26(49.1)	50(45.1)/61(55.0)	0.233 ^2^
Alcohol (unit ^3^/week)	1.54 ± 2.24	3.30 ± 4.51	2.49 ± 3.72	0.082
Smoking (Y/N)	0(0.0)/58(100.0)	2(3.8)/51(96.2)	2(1.8)/109(98.2)	0.226 ^4^
Smoking (cigarettes/day)	0.00 ± 0.00	12.5 ± 3.5	12.5 ± 3.5	-

Values are presented as mean ± standard deviation (SD) or number. ^1^ Analyzed using independent *t*-test. ^2^ Analyzed using chi-square test. ^3^ Alcohol 1 unit = Alcohol 10 g = Alcohol 12.5 mL. ^4^ Analyzed using Fisher’s exact test. Abbreviations: BMI, body mass index; SBP, systolic blood pressure; DBP, diastolic blood pressure; NK, natural killer; interleukin-2, 6, 12 (IL-2, IL-6, IL-12); INF-γ, interferon gamma; TNF-α, tumor necrosis factor alpha.

**Table 3 nutrients-12-01642-t003:** Intake of major nutrients and MET values in the PTE and placebo groups measured at baseline and at 8 weeks.

	PTE Group (*n* = 58)				Placebo Group (*n* = 53)				
	Baseline	Week 8	Change	*p*-Value ^1^	Baseline	Week 8	Change	*p*-Value ^1^	*p*-Value ^2^
Energy (kcal)	1538.2 ± 537.3	1552.5 ± 394.5	14.4 ± 501.1	0.828	1682.4 ± 568.5	1580.9 ± 520.4	−101.6 ± 525.5	0.165	0.237
Carbohydrates (g)	236.9 ± 67.4	264.0 ± 66.2	29.1 ± 69.0	0.002	251.9 ± 85.1	247.6 ± 81.3	−4.33 ± 79.6	0.694	0.020
Lipids (g)	40.3 ± 26.9	34.6 ± 18.9	−5.7 ± 28.8	0.137	46.4 ± 17.1	40.9 ± 23.9	−5.4 ± 31.9	0.219	0.965
Protein (g)	62.1 ± 27.1	52.9 ± 17.5	−9.1 ± 26.4	0.011	64.9 ± 23.2	61.7 ± 29.4	−3.2 ± 26.9	0.386	0.250
Fiber (g)	19.3 ± 7.9	21.5 ± 8.1	2.2 ± 9.0	0.067	19.8 ± 7.7	21.4 ± 9.7	1.6 ± 8.2	0.169	0.697
MET (min/week)	1886.9 ± 2055.2	3016.6 ± 5722.3	1149.7 ± 5732.8.3	0.132	2891.7 ± 4699.8	2307.9 ± 3518.7	−583.8 ± 4403.9	0.339	0.079

Values are presented as mean ± standard deviation (SD). ^1^ Analyzed using paired *t*-test. ^2^ Analyzed using linear mixed model between groups. Abbreviations: MET, metabolic equivalent; PTE, *Porphyra tenera* extract.

**Table 4 nutrients-12-01642-t004:** NK cell activity and cytokine cluster analysis obtained prior to and after treatment in the two groups.

	PTE Group (*n* = 58)				Placebo Group (*n* = 53)					*Adj.*
	Baseline	Week 8	Change	*p*-Value ^1^	Baseline	Week 8	Change	*p*-Value ^1^	*p*-Value ^2^	*p*-Value ^4^
NK cell activity (%)										
12.5:1	11.4 ± 8.4	14.8 ± 8.2	3.4 ± 6.9	0.0004	11.8 ± 9.1	13.3 ± 9.6	1.6 ± 6.3	0.079	0.1390	0.0829
25:1	20.6 ± 12.2	23.6 ± 12.8	3.0 ± 7.4	0.0034	20.7 ± 13.6	21.1 ± 13.1	0.5 ± 7.0	0.638	0.0692	0.0587
50:1	28.0 ± 14.3	30.8 ± 14.6	2.8 ± 7.3	0.0055	27.4 ± 14.3	28.1 ± 14.6	0.6 ± 8.1	0.585	0.1401	0.0832
IL-2 (pg/mL)	1.28 ± 0.85	1.28 ± 0.92	0.00 ± 0.45	0.9909	1.44 ± 1.51	1.37 ± 1.06	−0.08 ± 1.02	0.5792	0.6054	0.7319
IL-6 (pg/mL)	0.64 ± 0.69	0.69 ± 0.89	0.06 ± 0.79	0.5922	0.63 ± 0.85	0.53 ± 0.51	−0.10 ± 0.50	0.1384	0.2022	0.2582
IL-12 (pg/mL)	2.31 ± 1.86	2.18 ± 1.32	−0.13 ± 0.78	0.2007	2.08 ± 1.49	1.89 ± 1.05	−0.19 ± 0.87	0.1245	0.7270	0.8367
INF-γ (pg/mL)	8.35 ± 8.68	8.00 ± 6.75	−0.35 ± 2.59	0.3135	7.04 ± 6.97	6.41 ± 4.49	−0.63 ± 3.53	0.2029	0.6378	0.6044
TNF-α (pg/mL)	6.05 ± 2.47	7.01 ± 3.53	0.96 ± 3.76	0.0577	7.36 ± 2.82	7.94 ± 4.78	0.58 ± 5.48	0.4420	0.6798	0.3650 0.4097 ^3^

Values are presented as mean ± standard deviation (SD). ^1^ Analyzed using paired *t*-test. ^2^ Analyzed using linear mixed model between groups. ^3^ Analyzed using ANCOVA after adjusting for baseline values. ^4^ Analyzed using ANCOVA after adjusting for carbohydrate and T-MET change values. Abbreviations: MET, metabolic equivalents; NK, natural killer; PTE, *Porphyra tenera* extract; IL-2, IL-6, IL-12, interleukin-2, 6, 12; IFN-γ, interferon gamma; TNF-α, tumor necrosis factor alpha.

**Table 5 nutrients-12-01642-t005:** NK cell activity and cytokine cluster analysis obtained prior to and after treatment in the two subgroups.

	PTE Group (*n* = 50)			Placebo Group (*n* = 51)					
	Baseline	Week 8	Change	*p*-Value ^1^	Baseline	Week 8	Change	*p*-Value ^1^	*p*-Value ^2^
NK cell activity (%)									
12.5:1	11.2 ± 8.4	14.6 ± 8.0	3.4 ± 7.3	0.0019	11.8 ± 9.2	12.8 ± 9.5	1.0 ± 5.7	0.2238	0.0683
25:1	20.3 ± 12.0	23.2 ± 12.4	2.8 ± 7.5	0.0106	20.6 ± 13.8	20.5 ± 13.0	−0.1 ± 6.4	0.8907	0.0357
50:1	27.8 ± 14.2	30.5 ± 14.4	2.7 ± 7.4	0.0134	27.3 ± 14.4	27.3 ± 14.5	0.1 ± 7.5	0.9457	0.0810
IL-2 (pg/mL)	1.33 ± 0.875	1.36 ± 0.94	0.03 ± 0.46	0.6064	1.44 ± 1.51	1.36 ± 1.07	−0.08 ± 1.04	0.5885	0.4811
IL-6 (pg/mL)	0.70 ± 0.72	0.65 ± 0.58	−0.05 ± 0.34	0.3395	0.64 ± 0.87	0.53 ± 0.52	−0.11 ± 0.51	0.1427	0.4824
IL-12 (pg/mL)	2.51 ± 1.92	2.34 ± 1.34	−0.17 ± 0.83	0.1451	2.10 ± 1.51	1.91 ± 1.06	−0.19 ± 0.89	0.1263	0.9097
INF-γ (pg/mL)	8.95 ± 9.18	8.56 ± 7.06	−0.39 ± 2.74	0.3175	7.13 ± 7.08	6.50 ± 4.55	−0.63 ± 3.60	0.2171	0.7084
TNF-α (pg/mL)	6.00 ± 2.56	7.09 ± 3.60	1.13 ± 3.72	0.0368	7.31 ± 2.86	7.59 ± 3.83	0.29 ± 4.65	0.6606	0.3180 0.7997 ^3^

Values are presented as mean ± standard deviation (SD). ^1^ Analyzed using paired *t*-test. ^2^ Analyzed using linear mixed model between groups. ^3^ Analyzed using ANCOVA after adjustment for baseline values Abbreviations: MET, metabolic equivalents value; NK, natural killer; PTE, *Porphyra tenera* extract; IL-2, IL-6, IL-12, interleukin-2, 6, 12; IFN-γ, interferon gamma; TNF-α, tumor necrosis factor alpha.

## Data Availability

The datasets generated and/or analyzed during the current study are not publicly available to protect patient confidentiality but are available from the corresponding author on reasonable request.

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
