# Peer review of "Effects of Porphyra tenera Supplementation on the Immune System: A Randomized, Double-Blind, and Placebo-Controlled Clinical Trial"

_nutrients, 2020, doi:10.3390/nu12061642_

Round 1

Reviewer 1 Report

The purpose of this clinical study was to investigate the safety profile and immunomodulatory properties of Porphyra tenera extract (PTE) on healthy volunteers. The significance of this study lays in the fact that Porphyra tenera is a staple food in Western Pacific Ocean regions and Eastern Asia, while no other human studies have been conducted about its beneficial actions on the consumer. However, several points made throughout the manuscript require our attention.

  1. Although PTE consumption did not seem to have adverse effects, no beneficial effects were observed, as well.
  2. Τhe authors concluded that: PTE supplementation for 8 weeks increased NK-cell activity (%) compared to placebo (line 418) and: these results suggest that 8 weeks of PTE supplementation can improve immune function through improving NK-cell activity in healthy people (lines 424-425). However, this increase did not reach statistical significance (p<0.1).
  3. The results from the URI study were not conclusive, as the sample was very small (PTE group=8 participants, placebo group=2 participants). The authors did acknowledge this limitation (lines 409-410), however, they stated that: Total score variation of URI symptoms was significantly lower in the PTE group than the placebo group. This confirms that PTE supplementation can aid in rapid recovery from URI symptoms (lines 420-422). The investigation of the actions of PTE against URI should be conducted in a separate study that would take into consideration more parameters, and of course, be of a larger scale.
  4. The authors stated that: Among subjects with a URI, URI symptom total score in the PTE group (n=8) decreased from 3.00±2.16 at baseline to 1.50±0.71 after 8 weeks of intake, but the score in the placebo group (n=2) increased from 3.00±0.0 at baseline to 4.00±0.0 after 8 weeks of intake (lines 275-278). Did the authors expect URI symptoms to last for 8 weeks? URI symptoms typically last 2–14 days, and most people recover in about 7–10 days unless there are some underlying conditions that prolong recovery.

Other points that require attention in the manuscript are:

  • Lines 90-107: the exclusion criteria could be better presented in a table
  • Line 109: the word ‘random’ should be replaced with ‘randomized’
  • Lines 166-175: When did these evaluations take place? (0, 4 and/or 8 weeks)
  • Lines 384-386: the authors did not test the effect of age in the efficacy of PTE supplementation and there is no data to support their claim ‘This suggests that age moderates the effects of PTE and related supplements on cytokine levels.'

Author Response

Reviewer report

Reviewer 1

The purpose of this clinical study was to investigate the safety profile and immunomodulatory properties of Porphyra tenera extract (PTE) on healthy volunteers. The significance of this study lays in the fact that Porphyra tenera is a staple food in Western Pacific Ocean regions and Eastern Asia, while no other human studies have been conducted about its beneficial actions on the consumer. However, several points made throughout the manuscript require our attention.

Q1. Although PTE consumption did not seem to have adverse effects, no beneficial effects were observed, as well.

Response: I agree with your advice.

However, in the present study, subgroup analysis of subjects without a URI was performed to exclude the effects of viral infection on NK cell activity. In subgroup analysis of subjects without a URI (n = 101) and of all subjects (n = 111), NK cell activity tended to increase in the PTE group for all E:T ratios (E:T = 12.5:1, E:T = 25:1, and E:T = 50:1; p < 0.10) compared with the placebo group; however, significant differences were not observed between the two groups. Subjects without a URI in the PTE group showed increased NK cell activity at an E:T ratio of 25.1:1 compared with the placebo group. In several previous studies (1-7) enhanced NK cell activity in healthy adults was reported only at certain E:T ratios; in the present study, enhanced NK cell activity was consistently identified in the PTE group at all E:T ratios (E:T = 12.5:1, E:T = 25:1, and E:T = 50:1). These results suggested that of PTE supplementation is effective for enhancing immune activity.

References

  1. Lee, Y.J.; Paik, D.-J.; Kwon, D.Y.; Yang, H.J.; Park, Y. Agrobacterium sp.-derived β-1, 3-glucan enhances natural killer cell activity in healthy adults: a randomized, double-blind, placebo-controlled, parallel-group study. Nutrition research and practice 2017, 11, 43-50.
  2. Kwak, J.H.; Baek, S.H.; Woo, Y.; Han, J.K.; Kim, B.G.; Kim, O.Y.; Lee, J.H. Beneficial immunostimulatory effect of short-term Chlorella supplementation: enhancement of Natural Killercell activity and early inflammatory response (Randomized, double-blinded, placebo-controlled trial). Nutrition journal 2012, 11, 53.
  3. Choi, J.-y.; Paik, D.-J.; Kwon, D.Y.; Park, Y. Dietary supplementation with rice bran fermented with Lentinus edodes increases interferon-γ activity without causing adverse effects: a randomized, double-blind, placebo-controlled, parallel-group study. Nutrition journal 2014, 13, 35.
  4. Hwang, J.-T.; Cho, J.M.; Jeong, I.H.; Lee, J.-y.; Ha, K.-C.; Baek, H.-I.; Yang, H.J.; Kim, M.J.; Lee, J.H. The effect of silk peptide on immune system, A randomized, double-blind, placebo-controlled clinical trial. Journal of functional foods 2019, 55, 275-284.
  5. Jung, S.J.; Hwang, J.H.; Oh, M.R.; Chae, S.W. Effects of Cordyceps militaris supplementation on the immune response and upper respiratory infection in healthy adults: a randomized, double-blind, placebo-controlled study. Journal of Nutrition and Health 2019, 52, 258-267.
  6. Jung, S.-J.; Jung, E.-S.; Choi, E.-K.; Sin, H.-S.; Ha, K.-C.; Chae, S.-W. Immunomodulatory effects of a mycelium extract of Cordyceps (Paecilomyces hepiali; CBG-CS-2): a randomized and double-blind clinical trial. BMC complementary and alternative medicine 2019, 19, 77.
  7. Nakagami, Y.; Suzuki, S.; Espinoza, J.L.; Vu Quang, L.; Enomoto, M.; Takasugi, S.; Nakamura, A.; Nakayama, T.; Tani, H.; Hanamura, I. Immunomodulatory and Metabolic Changes after Gnetin-C Supplementation in Humans. Nutrients 2019, 11, 1403. 

Q2. Τhe authors concluded that: PTE supplementation for 8 weeks increased NK-cell activity (%) compared to placebo (line 418) and: these results suggest that 8 weeks of PTE supplementation can improve immune function through improving NK-cell activity in healthy people (lines 424-425). However, this increase did not reach statistical significance (p<0.1).

Response: As you suggested, the contents of the conclusion were revised and presented. (Lines 395-399, page 18).

Q3. The results from the URI study were not conclusive, as the sample was very small (PTE group=8 participants, placebo group=2 participants). The authors did acknowledge this limitation (lines 409-410), however, they stated that: Total score variation of URI symptoms was significantly lower in the PTE group than the placebo group. This confirms that PTE supplementation can aid in rapid recovery from URI symptoms (lines 420-422). The investigation of the actions of PTE against URI should be conducted in a separate study that would take into consideration more parameters, and of course, be of a larger scale.

Response: I agree with your advice. I will delete the (409~410 and 420-422)line, which is the URI symptoms. Not only the number of subjects but also the change in URI symptom scores was excluded from the discussion because the statistical significant differences were not confirmed.

Q4. The authors stated that: Among subjects with a URI, URI symptom total score in the PTE group (n=8) decreased from 3.00±2.16 at baseline to 1.50±0.71 after 8 weeks of intake, but the score in the placebo group (n=2) increased from 3.00±0.0 at baseline to 4.00±0.0 after 8 weeks of intake (lines 275-278). Did the authors expect URI symptoms to last for 8 weeks? URI symptoms typically last 2–14 days, and most people recover in about 7–10 days unless there are some underlying conditions that prolong recovery. 

Response: I agree with your advice. In this study, URI objects are very small sample sizes. We will exclude it from the research results as discussed.

According to several previous studies, the intake of Echinacea pura extract [1] over four months confirmed a significant decrease in the incidence of colds and the number of days of colds, and the total intake of 600 mg of bovine lactoferrin and Ig-rich fraction (Lf /IgF) compounds [2] per day decreased in total incidence of reduction and accumulation of the incidence of reduction in the incidence of the incidence. Therefore, during the 8 weeks of participation in the study, this researcher was investigated for additional observation of changes in URI incidence and URI symptom scores for all subjects. 

References:

  1. Jawad M, Schoop R, Suter A, Klein P, Eccles R. Safety and efficacy profile of Echinacea purpurea to prevent common cold episodes: A randomized, double-blind, placebo-controlled trial. Evid Based Complement Alternat Med 2012; 2012: 1-8.
  2. Vitetta L, Coulson S, Beck SL, Gramotnev H, Du S, Lewis S. The clinical efficacy of a bovine lactoferrin/whey protein Ig-rich fraction (Lf/IgF) for the common cold: a double blind randomized study. Complement Ther Med 2013; 21(3):164-171.

Q5. Other points that require attention in the manuscript are:

Q 5-1 Lines 90-107: the exclusion criteria could be better presented in a table

Response: I tried to present it as a table as you suggested. However, since the number of tables is 5 or more, it seems to be necessary to present them as contents.

Please understand this situation.

Q 5-2 Line 109: the word ‘random’ should be replaced with ‘randomized’

Response: Thank you for the careful review.

          As you suggested, we made following changes (Line 111, page 4). 

Q 5-3 Lines 166-175: When did these evaluations take place? (0, 4 and/or 8 weeks)

Response: As you suggested, We have described the time of measurements of the safety outcomes (Line 168~169, page 5).

-> All subjects underwent safety evaluations at baseline (week 0) and after completing the 8weeks study.

Q 5-4 Lines 384-386: the authors did not test the effect of age in the efficacy of PTE supplementation and there is no data to support their claim ‘This suggests that age moderates the effects of PTE and related supplements on cytokine levels.’ 

Response: I agree with your advice. I will delete the 384-386 line, which is the part. We will not include what has been discussed without evaluation the effects of PTE on supplemental effects by age.

Thank you!

Reviewer 2 Report

The authors present an interesting study investigating the effect of PTE on the immune system. The manuscript is generally well presented however, due to the lack of structured paragraph in the results section and discussion makes quite cumbersome to read. I therefore, suggest English editing prior to final acceptance.

A major flaw in this study is the effect of PTE on upper respiratory infection. The incidence of upper respiratory infections would have been an incidental and therefore and random variable and cannot be used as an outcome.

Specific comment outlined below

Abstract

(line 12), ….“Subjects (3x103 ≤ peripheral blood”.. should be ≥.

Materials and methods

Section 2.2, selection criteria 1, Why the participants age was more than 50 years

Section 2.2, exclusion criteria 10 is not clear.

Section 2.4.3, secondary outcomes, no details provided on how URI was assessed, as URI was not induced in subjects, it should not represent an endpoint.

Results

Table 2, general demographic information should only include the final analysed data from 111 subjects.

Author Response

Reviewer report

Reviewer  2

Q1. Comments and Suggestions for Authors

The authors present an interesting study investigating the effect of PTE on the immune system. The manuscript is generally well presented however, due to the lack of structured paragraph in the results section and discussion makes quite cumbersome to read. I therefore, suggest English editing prior to final acceptance.

Response: Thank you for the suggestion. We have been used the service of English editors.  =>Attached files edited documents.

Q2. A major flaw in this study is the effect of PTE on upper respiratory infection. The incidence of upper respiratory infections would have been an incidental and therefore and random variable and cannot be used as an outcome. 

Response: As you suggested, URI's outcomes are not to be presented 

Q3. Specific comment outlined below

â—¾Abstract

Q3-1.(line 12), ….“Subjects (3x103 ≤ peripheral blood leukocyte levels≥”.. should be ≥.

Response: Thank you for the careful review.

           As you suggested, we made following changes (Line 15, page 1).

  • “Subjects (3 x103 ≤ peripheral blood leukocyte levels≥8 x103”.cell/㎕)

â—¾Materials and methods

Q3-2. Section 2.2, selection criteria 1, Why the participants age was more than 50 years

Response: Generally, the selection criteria are 25 to 70 years old [1-2]. However, in the recently published paper, males and females older than 50 years, were selected based on the criteria [3]. Also, Marie-Chantal Farges et. al.[4] Immune response in healthy human subjects is mostly affected by age rather than by dietary carotenoid depletion and repletion. Even in carefully selected healthy adults, some age-related immune changes occur predominantly from 50 years onwards. The subjects of this study selected the age of 50 years or older that shows a decrease in immunity.

References;

  1. Lee, Y.J.; Paik, D.-J.; Kwon, D.Y.; Yang, H.J.; Park, Y. Agrobacterium sp.-derived β-1, 3-glucan enhances natural killer cell activity in healthy adults: a randomized, double-blind, placebo-controlled, parallel-group study. Nutrition research and practice 2017, 11, 43-50.
  2. Choi, J.-y.; Paik, D.-J.; Kwon, D.Y.; Park, Y. Dietary supplementation with rice bran fermented with Lentinus edodes increases interferon-γ activity without causing adverse effects: a randomized, double-blind, placebo-controlled, parallel-group study. Nutrition journal 2014, 13, 35.
  3. Hwang, J.-T.; Cho, J.M.; Jeong, I.H.; Lee, J.-y.; Ha, K.-C.; Baek, H.-I.; Yang, H.J.; Kim, M.J.; Lee, J.H. The effect of silk peptide on immune system, A randomized, double-blind, placebo-controlled clinical trial. Journal of functional foods 2019, 55, 275-284.
  4. Marie-Chantal Farges, Re´gine Minet-Quinard1, Ste´phane Walrand1, Emilie Thivat, Josep Ribalta, Brigitte Winklhofer-Roob, Edmond Rock and Marie-Paule Vasson. Immune status is more affected by age than by carotenoid depletion–repletion in healthy human subjects. British Journal of Nutrition (2012), 108, 2054–2065

Q3-3. Section 2.2, exclusion criteria 10 is not clear.

Response: As you suggested, we made following changes (Line 107, page 3).

10) Those who were fertile and not taking contraceptives 

Q3-4. Section 2.4.3, secondary outcomes, no details provided on how URI was assessed, as URI was not induced in subjects, it should not represent an endpoint. 

Response: I agree with your advice. In this study, URI objects are very small sample sizes. We will exclude it from the research results as discussed.

â—¾Results

Q3-5.Table 2, general demographic information should only include the final analyzed data from 111 subjects. 

Response: As you suggested, we made following statistical analyzed data (revised Table 2)

Thank you!

Round 2

Reviewer 1 Report

The authors have addressed my comments and the resubmitted manuscript has been improved sufficiently for publication. Please check figure 1, it looks blurry.

Reviewer 2 Report

Thank for providing an updated version of the manuscript. I have no further comment.